# Inhibition of InsP3R with Xestospongin B Reduces Mitochondrial Respiration and Induces Selective Cell Death in T Cell Acute Lymphoblastic Leukemia Cells

**DOI:** 10.3390/ijms22020651

**Published:** 2021-01-11

**Authors:** Pablo Cruz, Ulises Ahumada-Castro, Galdo Bustos, Jordi Molgó, Daniela Sauma, Alenka Lovy, César Cárdenas

**Affiliations:** 1Center for Integrative Biology, Faculty of Sciences, Universidad Mayor, Santiago 8580745, Chile; pcruznunez@gmail.com (P.C.); uj.ahumada@gmail.com (U.A.-C.); galdo.bustos.c@gmail.com (G.B.); alenkalovy@gmail.com (A.L.); 2Geroscience Center for Brain Health and Metabolism, Santiago 8580745, Chile; 3Département Médicaments et Technologies pour la Santé, Service d’Ingénierie Moléculaire pour la Santé (SIMoS), ERL CNRS n° 9004, Institut des Sciences du Vivant Frédéric Joliot, CEA, Université Paris-Saclay, bâtiment 152, Point courrier 24, F-91191 Gif sur Yvette, France; jordi.molgo@cea.fr; 4Biology Department, Faculty of Science, Universidad de Chile, Santiago 8330015, Chile; dsauma@uchile.cl; 5Department of Neuroscience, Center for Neuroscience Research, Tufts University School of Medicine, Boston, MA 02111, USA; 6The Buck Institute for Research on Aging, Novato, CA 94945, USA; 7Department of Chemistry and Biochemistry, University of California, Santa Barbara, CA 92521, USA

**Keywords:** calcium, cancer, metabolism, bioenergetics, T-ALL

## Abstract

T-cell acute lymphoblastic leukemia (T-ALL) is an aggressive hematological malignancy whose chemoresistance and relapse persist as a problem despite significant advances in its chemotherapeutic treatments. Mitochondrial metabolism has emerged as an interesting therapeutic target given its essential role in maintaining bioenergetic and metabolic homeostasis. T-ALL cells are characterized by high levels of mitochondrial respiration, making them suitable for this type of intervention. Mitochondrial function is sustained by a constitutive transfer of calcium from the endoplasmic reticulum to mitochondria through the inositol 1,4,5-trisphosphate receptor (InsP3R), making T-ALL cells vulnerable to its inhibition. Here, we determine the bioenergetic profile of the T-ALL cell lines CCRF-CEM and Jurkat and evaluate their sensitivity to InsP3R inhibition with the specific inhibitor, Xestospongin B (XeB). Our results show that T-ALL cell lines exhibit higher mitochondrial respiration than non-malignant cells, which is blunted by the inhibition of the InsP3R. Prolonged treatment with XeB causes T-ALL cell death without affecting the normal counterpart. Moreover, the combination of XeB and glucocorticoids significantly enhanced cell death in the CCRF-CEM cells. The inhibition of InsP3R with XeB rises as a potential therapeutic alternative for the treatment of T-ALL.

## 1. Introduction

Acute T-lymphoblastic leukemia (T-ALL) is a rapidly proliferating hematological malignancy that accounts for 15% of childhood and 25% of adult Acute Lymphoblastic Leukemia (ALL) cases [1]. Although current glucocorticoid-based chemotherapy treatments [2,3] are remarkably successful and cure nearly 80% of pediatric and 50% of adult cases [4], a high percentage of primary and secondary resistance [5] still exists, which has encouraged the search for new agents and strategies to treat this disease. Small molecule inhibitors such as venetoclax (Bcl-2 inhibitor) [6,7], dasatinib (tyrosine kinase inhibitor) [8] and palbociclib (cyclin-dependent kinases 4 and 6 inhibitor) [9] have been used with some success. Immunotherapeutic approaches with the chimeric antigen receptor (CAR)-T [10,11] and monoclonal antibodies [12] have also shown encouraging results. Despite these efforts, no new agents for the treatment of relapsed and/or refractory T-ALL in adults and children have been approved by the FDA since nelarabine in 2005. Understanding the unique features of these cells is necessary to bring new options for treatment.

A hallmark feature of cancer cells is the reprogramming of their metabolic fitness, even when nutrients are available [13,14]. Warburg suggested that cancer originates from an irreversible injury to mitochondria followed by a compensatory increase in glycolysis [15], but increasing evidence indicates that many cancer cells rely on mitochondrial metabolism and use a significant fraction of glucose-derived pyruvate for ATP generation through oxidative phosphorylation (OXPHOS) [14,16,17]. It is important to mention that current reports indicate that metabolism reprograming observed in cancer cells depends on the tumor context and can lean toward glycolysis as first described by Warburg, or toward OXPHOS as observed more recently in many tumor types and was recently reviewed [18]. Emerging evidence from patient-derived tumors, which includes some forms of refractory leukemia such as AML, indicates that OXPHOS plays a main role in tumor cell survival by providing not only ATP, but essential metabolic intermediates for the synthesis of biomolecules [19]. In fact, T-ALL cells reprogram their metabolism and exhibit a limited metabolic flexibility compared to normal T lymphocytes, which can be characterized by a robust mitochondrial function driven by oncogenic NOTCH1 signaling, and activation of c-MYC [20,21,22].

Mitochondrial function is sustained by a constant uptake of Ca^2+^ constitutively released from the endoplasmic reticulum through the inositol 1,4,5-trisphosphate receptor (InsP3R) [23]. Interruption of this Ca^2+^ transfer to mitochondria generates a bioenergetic crisis characterized by the activation of AMPK and autophagy [23]. If unresolved, this Ca^2+^ interruption induces cell death in breast and prostate cancer cells without affecting normal cells [24]. Moreover, the transfer of Ca^2+^ to mitochondria through the InsP3R is necessary even in cancer cells where OXPHOS is not supported, because Ca^2+^ controls the activity of α-ketoglutarate dehydrogenase, which is key in these cells to generate the NADH needed to sustain reductive carboxylation [25].

In T-ALL cells, but not in normal T cells, the expression of ORP4L, a scaffold protein that allows the assembly of CD3ε, Gαq/11, and PLCβ3 into a complex facilitates the activation of PLCβ3 and the generation of inositol 1,4,5 trisphosphate (IP3), the substrate of InsP3R [26]. The knockdown of ORP4L suppresses PLCβ3 activity, generating a bioenergetic crisis that induces cell death [26]. These results confirm the importance of InsP3R Ca^2+^ signaling to maintain bioenergetics in T-ALL cells. Xestospongin B (XeB), a macrocyclic bis-1-oxaquinolizidine alkaloid extracted from the marine sponge *Xestospongia exigua* [27], is a highly specific inhibitor of the InsP3R [28] that interferes with the flow of InsP3R-Ca^2+^ released in breast and prostate cancer cells, causing a bioenergetic drop that induces cell death [23]. Here, we aim to determine whether XeB can selectively affect the bioenergetics and cell viability of the glucocorticoid-sensitive human T-ALL cell line CCRF-CEM and the glucocorticoid-insensitive human T-ALL Jurkat cell line. Our results show that XeB can selectively affect the viability of T-ALL cells and has the potential to become a therapeutic alternative.

## 2. Results

In agreement with previous reports [22,26], we found that glucocorticoid-sensitive human T-ALL cell line CCRF-CEM and the glucocorticoid-insensitive human T-ALL Jurkat cell line present high mitochondrial oxidative metabolism compared with B and T normal cells, as demonstrated by the Oxygen Consumption Rate (OCR) (Figure 1A). Analysis of the data in Figure 1 shows that CCRF-CEM and the Jurkat cells have higher basal OCR compared with B and T normal cells (Figure 1B). However, maximal respiration, as well as the ability of B and T cells to increase their respiration upon stress, a phenomenon known as spare respiratory capacity, was significantly higher in these cells compared to CCRF-CEM and Jurkat cells (Figure 1C,D). The OCR link to ATP production shows that the high levels of basal oxygen consumption in CCRF-CEM and Jurkat cells are used to generate ATP (Figure 1E). Altogether, these results show that T-ALL cell lines CCRF-CEM and Jurkat are using mitochondria to the maximum of their capability to generate ATP and maintain their homeostasis.

In addition, we evaluated the basal energetic status of these cells by analyzing the activation of the AMP-activated protein kinase (AMPK) by detecting the phosphorylation in its T172 residue. We observed that in T-ALL cells, AMPK shows a tendency to be phosphorylated, although no statistical significance was reached. This suggests that AMPK is active in basal conditions compared with normal T cells. This suggests that despite the increased basal and ATP-linked OCR, T-ALL cells might be experiencing a metabolic/energetic stress (Figure 1F) in agreement with a previous report, which indicates a similar basal increasing of pT172-AMPK/AMPK ratio [22]. Next, we compared glycolytic function in these cells by determining the extracellular acidification rate (ECAR) (Figure 2A). Jurkat cells exhibit the highest glycolysis, while CCRF-CEM, B, and T cells show similar glycolysis levels (Figure 2B). When we calculated the glycolytic reserve capacity, which is the capability of these cells to increase glycolysis upon a challenge, we found that CCRF-CEM and B cells exhibited the highest glycolytic reserve, while Jurkat cells showed low levels (Figure 2C). On the other hand, T cells show no glycolytic reserve capacity. In summary, our results in CCRF-CEM and Jurkat cells confirm previous observations in T-ALL cells regarding their glycolytic capacity and the high use of oxidative phosphorylation (OXPHOS) [22].

Previously, we have shown that constitutive InsP3R-mediated Ca^2+^ transfer to mitochondria is essential to maintain cellular bioenergetics in several cell types [23]. How the inhibition of this Ca^2+^ transfer to mitochondria affects cellular bioenergetics in T-ALL, B and T cells has not yet been described. To this end, we first determined whether 5 µM of the specific InsP3R inhibitor Xestospongin B (XeB) was able to inhibit InsP3R-evoked Ca^2+^ transients in CCRF-CEM, Jurkat, and normal T cells. As shown in Figure 3, XeB almost completely blocked the InsP3R-mediated Ca^2+^ increase induced by ATP in all cell lines. Then, we treated CCRF-CEM, Jurkat, B, and T cells with 5 µM of XeB for 1 h and oxygen consumption was evaluated. Basal respiration in CCRF-CEM cells decreased from 99.5 ± 4.1 to 78.4 ± 3.9 pmol/min/µg protein, while in Jurkat cells it decreased from 106.5 ± 6.2 to 60.2 ± 4.9 pmol/min/µg protein (Figure 4A,B). Surprisingly, no effect was observed in B and T cells (Figure 4C,D). The maximal respiration exhibited the same trend with a decrease from 117.2 ± 3.3 to 95.6 ± 2.1 pmol/min/µg protein in CCRF-CEM and from 78.9 ± 3.9 to 60.53 ± 3.3 pmol/min/µg protein in Jurkat cells (Figure 4A,B). Interestingly, upon InsP3R inhibition in B and T cells, we also observed a decrease in the maximal respiration from 146.03 ± 3.2 to 122.7 ± 3.6 pmol/min/µg protein in B cells and from 156.3 ± 4.8 to 118.6 ± 3.9 pmol/min/µg protein in T cells (Figure 4C,D). Overall, the respiration data indicate that the inhibition of InsP3R with XeB induces a bioenergetic stress in all cell lines, as we have observed before [23,24]. To confirm this, we determined the redox and energetic status in CCRF-CEM, Jurkat, and normal T cells, indicated by the NAD^+^/NADH ratio, upon XeB treatment. We observed that the pharmacological inhibition of InsP3R increases the NAD^+^/NADH in T-ALL cells and to a lesser extent, in their normal counterpart (Figure 5). This increase in the NAD^+^/NADH ratio reflects a decrease in mitochondrial function as we have demonstrated previously [25,29]. In physiological conditions, active T cells play an important antitumor role, and disturbing their normal homeostasis could have detrimental effects. We therefore determined whether XeB affects normal bioenergetics in activated T cells. T cells were activated with CD3 and CD28. Once activated, T cells were treated with XeB, and the bioenergetics profile was assayed. Activated T cells exhibited an increase in basal and maximum respiration compared with the naïve counterpart, as previously described [30,31,32]. Similar to what we have observed in naïve B and T cells, the presence of XeB did not affect basal respiration, but decreased the maximal respiration rate in activated T cells (Figure 6).

We have previously shown that the bioenergetic crisis induced by interruption of Ca^2+^ transfer from the endoplasmic reticulum to mitochondria with XeB induces cell death in breast and prostate cancer cells, without affecting normal cells [24] through a mechanism involving a decrease in the active isoform of PDH [25]. More recently, we have discovered that this mechanism also acts through the modulation of αKGDH activity and the decrease in OXPHOS and reductive carboxylation [25]. Since T-ALL cells utilize mitochondria to a high extent, we evaluated whether XeB induces cell death in CCRF-CEM and Jurkat cells in a selective fashion. As shown in Figure 7A, 24 h treatment with 5 µM XeB induces a significant increase in cell death in Jurkat cells (~40%) and CCRF-CEM cells (~20%), without affecting B and T cells. At 48 h treatment, we also observed that cell death increased, reaching approximately 70% in Jurkat cells and 50% in CCRF-CEM. Once again, no effect on cell viability was observed in B and T cells, suggesting a selective effect on cancer cells.

Reduction in mitochondrial function causes a decline in energy supply, but more importantly, a shortage of metabolic intermediates for the synthesis of lipids, proteins, DNA, and RNA [33]. Along this line, we have shown that the effect of InsP3R-mediated Ca^2+^ transfer to mitochondria inhibition can be overcome either by boosting mitochondrial function with metabolic intermediates, such as α-ketoglutarate [25], or by supplying nucleosides for the generation of necessary nucleotides [24]. Here, we aim to determine whether the same effect is observed in T-ALL cells. Therefore, we incubated CCRF-CEM and Jurkat cells with 5 µM XeB for 48 h in the presence of α-ketoglutarate and nucleosides, and cell death was determined with flow cytometry. Surprisingly, no protection was observed (Figure 7B). In addition, we explored whether the presence of non-essential amino acids (NEAA) would be able to reduce the effects of XeB, but this was also ineffective (Figure 7C). These results suggest that metabolic flexibility and plasticity might be reduced in T-ALL cells, which leads to cell death.

Finally, we sought to determine whether XeB would interfere with the effect of dexamethasone, a glucocorticoid broadly used to treat T-ALL patients. Surprisingly, the combination of both drugs shows a dramatically increased effect. XeB and dexamethasone alone have similar effects, inducing roughly 25% cell death in 24 h, but in combination, they cause 80% cell death in the CCRF-CEM cell line (Figure 8).

## 3. Discussion

Currently, it is widely accepted that cancer cells present higher energy productive phenotypes than their normal counterparts [34,35,36]. Particularly, each cancer type exhibits a unique metabolic profile with a variable contribution of either glycolysis or OXPHOS to the energetic budget [37]. In this sense, either OXPHOS or glycolysis could be a reliable selective target for the treatment of cancer cells, as has been broadly described before [38,39,40,41,42]. For this reason, characterization of the main metabolic routes in each cancer type is a pivotal step for an accurate identification of potential vulnerabilities, and the subsequent pharmacological design of cancer treatment. In this study, we determined that OXPHOS is consistently enhanced in T-ALL cell lines, which makes it a suitable therapeutic target. Increases in OXPHOS have been described in other hematological malignancies, such as acute myelogenous leukemia (AML) [43], chronic lymphocyte leukemia (CLL) [44], and diffuse large B cell lymphoma [45], as well as many other types of cancer such as pancreatic [46], lung adenocarcinoma [47], endometrial carcinoma [47], melanoma [48,49], and glioma [47], which make the discovery and development of inhibitors that affect OXPHOS function, such as XeB, an essential task.

Glycolysis has been described to be higher in T-ALL cells compared with naïve T cells [22]. In particular, Jurkat and CCRF-CEM cells exhibit a high energetic profile characterized by high glycolysis and high OXPHOS [50], as we determined in this study. Normal naïve T cells are expected to have high OXPHOS and low glycolysis; however, in our study, glycolytic flux in these cells was high and the glycolytic reserve capacity was low. Here, it is important to mention that CO_2_ generated during respiration also contributes to the acidification of the media and influences the extracellular acification rate (ECAR) measurement. Depending on the cell type, CO_2_ contribution to ECAR can range from 0% to 100% [51]. As we did not control for CO_2_ acidification, we cannot exclude that respiration contributes to our ECAR measurement, and in particular, to the high glycolytic flux observed in normal naïve T cells. Additional glycolysis measurements will be necessary in future studies to accurately evaluate glycolysis in these cell lines. On the other hand, it is possible that this metabolic behavior corresponds to a Crabtree effect, which describes that OXPHOS is inhibited by higher concentrations of glucose in the media, such as the one we used in our work (25 mM). High levels of glycolysis are normally observed in activated lymphocytes [52], reducing the therapeutic window of glycolysis as a cancer target. Accordingly, we believe that OXPHOS rather than glycolysis, represents a higher vulnerability to selectively treat T-ALL malignancy, although more studies are necessary to determine the real potential of this intervention.

In addition to their energetic role, it is broadly accepted that mitochondria play a central biosynthetic role in cancer [35,53] that sustains the proliferation requirement even under OXPHOS disruption [54]. The tricarboxylic acid (TCA) cycle, the “engine” of mitochondria, is sustained in cancer by a high uptake of the non-essential amino acid glutamine [55], which is converted to glutamate by glutaminases and subsequently is converted to α-ketoglutarate (α-KG) by either glutamate dehydrogenases or aminotransferases [56]. In the TCA cycle, α-KG is used as a substrate by the α-KG dehydrogenase (α-KGDH), a Ca^2+^-dependent rate-limiting enzyme [57]. In fact, the other rate-limiting enzymes in the TCA cycle, isocitrate dehydrogenase (ISDH) and pyruvate dehydrogenase, are also regulated by Ca^2+^ [58], as well as five other important proteins for mitochondrial function [59], suggesting that the presence of Ca^2+^ in the mitochondrial matrix plays a pivotal role in mitochondrial function. The transfer of Ca^2+^ to mitochondria occurs in special regions of contact between the endoplasmic reticulum and the mitochondria known as MAMs (mitochondria-associated membranes) and is mainly driven by the InsP3R [60,61]. The InsP3R is essential to maintain mitochondrial activity in normal and cancer cells and its inhibition causes a bioenergetic crisis that selectively kills cancer cells [23,24,25], as we observed here in T-ALL cells (Figure 9). In this context, our bioenergetic analyses indicate that malignant T-cells rely more on OXPHOS than their non-malignant counterpart, in agreement with previous reports [21,22], as shown by increased basal and ATP-linked OCR observed in CCRF-CEM and Jurkat cells (Figure 1). For this reason, we speculated a central role for InsP3R-mediated Ca^2+^ flux in the enhanced mitochondrial metabolism observed in T-ALL cells. This idea was supported by the inhibition of InsP3R-mediated Ca^2+^ by XeB, which leads the basal OCR of malignant cells to values close to their normal counterparts as shown in Figure 3. Additionally, the small spare respiratory capacity, the glycolytic reserve, basal AMPK activation, and increased NAD^+^/NADH ratio observed in malignant T cells suggest a limited metabolic flexibility and plasticity, which is in line with the previously described chronic metabolic stress observed in T-ALL cells [22]. Of note, as we did not perform a FCCP OCR dose–response curve for each cell line, it is possible that the spare respiratory capacity might be underestimated. Despite this, the main conclusion remains unchanged—InsP3R-mediated Ca^2+^ inhibition by XeB critically affects mitochondrial performance, causing selective cell death in T-ALL cells. The metabolic flexibility and plasticity are mainly mediated by AMPK, the “master regulator” of cell metabolism, which exerts an essential regulatory function on metabolic reprogramming [22]. This mechanism might explain the differences in OCR and ECAR values between T-ALL cells and CCRF-CEM and Jurkat cells. In this context, InsP3R activity seems to contribute to the coordination of cell metabolism [23]; however, how this channel would participate in the establishment of this long-term adaptability in T-ALL cells is unclear. The NAD^+^/NADH ratio increase observed after XeB (Figure 5) raises the possibility that deacetylases activated by NAD^+^ such as sirtuins participate or mediate the selective vulnerability of T-ALL cells. Additionally, increase in the NAD^+^/NADH ratio has been shown to diminish ROS production [62,63], whereas a decrease in ROS might occur in T-ALL treated with XeB and play a role in the cell death observed. However, a previous report shows that XeB does not change ROS generation in breast cancer cell lines [24]. In the same study, the treatment with the antioxidant N-acetyl-cysteine (NAC) does not restore colony formation. Nevertheless, as we have not conducted ROS measurements in T-ALL, we cannot exclude that in these cells, XeB could exert part of its effect through changes in ROS generation. More studies are necessary to explore the role of sirtuins and ROS as potential mechanisms involved in cell death. Moreover, since AMPK drives metabolic reprograming [22], activation of this master regulator should be limited in T-ALL malignancy or otherwise, its hyperactivity could contribute to cell death signaling as reported before in other blood cancer cells [64]. On the other hand, the increased cell death elicited by the inhibition of Ca^2+^ flux by XeB in the malignant T-ALL cells (Figure 7 and Figure 8) confirms that mitochondrial metabolism, enhanced by InsP3R activity, is selectively vulnerable in these malignant cells. Therefore, the use of the inhibitor XeB emerges as a novel and selective pharmacological approach to induce cell death in T-ALL cells (Figure 9).

As InsP3R-mediated Ca^2+^ flux to mitochondria is a potential vulnerability of T-ALL cells, it is necessary to explore the detailed mechanism by which Ca^2+^-dependent metabolic reprograming is limited in these malignant cells. In this sense, the selective vulnerability of malignant T-cells proposed in the present study is also supported by other reports. In line with our work, it has recently been described that T-ALL and leukemia stem cells highly express the oxysterol-binding protein (OSBP)-related protein 4 ORP4L, which acts as an adaptor/scaffold assembling CD3ε, Gαq/11, and PLCβ3 into a complex that activates PLCβ3. The activation of PLCβ3 catalyzes IP3 production, inducing InsP3R-mediated Ca^2+^ release necessary to maintain enhanced oxidative phosphorylation and mitochondrial function. Knockdown or pharmacological inhibition of ORP4L results in decreased InsP3R activity, which leads to suboptimal bioenergetics and subsequently, to a selective T-ALL and leukemia stem cell death in vitro and in vivo [26,65] such as we observed with the treatment with the inhibitor XeB. In addition, the lack of InsP3R-mediated Ca^2+^ can also impact transcription programs since its presence in the nucleus regulates CREB phosphorylation [66], which might limit the metabolic plasticity of cancer cells and consequently, the viability of these malignant cells. On the other hand, in chronic lymphocytic leukemia (CLL), large B-lymphoma cells, and multiple myeloma, the antiapoptotic protein Bcl-2 binds the InsP3R, suppressing its activity to prevent cytosolic Ca^2+^ increase-mediated apoptosis. The disruption of this interaction with a peptide that selectively targets the BH4 domain of Bcl-2 generates apoptosis and selective cell death [67,68,69]. This apparently opposite result shows the complexity of Ca^2+^ signaling and confirms that a tight regulation of Ca^2+^ kinetics, amplitude, and localization is fundamental to maintain cellular homeostasis.

Herein, we exploited T-ALL metabolic reprogramming and the ability of the InsP3R-mediated Ca^2+^ signal to modulate mitochondrial function to selectively jeopardize the viability of T-ALL cells. Our data show that mitochondrial Ca^2+^ modulators, such as XeB, are reliable and potential novel therapeutic tools to treat T-ALL malignancies. More studies are necessary to determine the mechanism that drives cell death in T-ALL cells after inhibition of Ca^2+^ transfer to the mitochondria.

## 4. Materials and Methods

### 4.1. Chemical and Reagents

All chemical and reagents were obtained from Sigma-Aldrich Corp. (Burlington, MA, USA). Stock solutions of all compounds were prepared in dimethyl sulfoxide (DMSO). XeB was provided by Dr. Jordi Molgó.

### 4.2. Cell Culture

CCRF-CEM (glucocorticoid-sensitive human T-ALL cell line) and Jurkat (glucocorticoid-insensitive human T-ALL cell line) were cultured in RPMI 1640 supplemented with 10% FBS, 100 U mL^−1^ penicillin, 100 µg mL^−1^ streptomycin, and 0.25 µg mL^−1^ fungizone (Gibco, Carlsbad, CA, USA) at 37 °C (95%/5% air/CO_2_).

### 4.3. Isolation of Peripheral Blood Mononuclear Cells

Peripheral blood mononuclear cells (PBMC) were isolated from the blood of healthy donors by density gradient centrifugation using Ficoll-Paque (GE Healthcare, Chicago, IL, USA). Gradients were centrifuged at 400× *g* for 30 min at room temperature in a swinging bucket rotor (without the brake applied). PBMC were incubated with fluorochrome-labeled anti-CD3 and anti-CD45R/B220 antibodies (Biolegend, San Diego, CA, USA). T cells (CD3+/B220−) and B cells (CD3−B220+) were isolated by cell sorting using a FACS Aria^TM^ III Cell Sorter (BD Biosciences, San José, CA, USA). Purified B and T cells were then incubated in RPMI + 10% FCS. This study was undertaken with ethical approval from “The Research Ethics Committee at the Hospital Clínico Universidad de Chile” and is in conformity with the provisions of the Declaration of Helsinki. All healthy donors gave their written informed consent before their blood was drawn

### 4.4. Activation of T Cells

Naive T cells purified from PMBC were incubated with CD3/CD28 antibodies at a concentration of 5 and 1.5 µg/mL, respectively, for 3 weeks. The antibodies were generously provided by Dr Patricia Luz (Universidad de los Andes).

### 4.5. Bioenergetic Profile

Oxygen consumption rates (OCRs) from naive B and T cells, Jurkat, and CCRF-CEM cells (0.1 × 10^5^ cells) were measured in non-buffered DMEM without phenol red (containing 25 mM glucose, 2 mM l-glutamine, and 1 mM sodium pyruvate) under basal conditions, and in response to 1 µM oligomycin, 0.5 µM fluoro-carbonyl cyanide phenylhydrazone (FCCP), and 1 µM rotenone + 1 µM antimycin A (Sigma-Aldrich, Burlington, MA, USA). Cells were attached using Poly-L-Lysine (Sigma-Aldrich Cat# P4707) pre-coated 96-plates. Extracellular acidification rate (ECAR) of naive B and T cells, Jurkat, and CCRF-CEM cells were measured in non-buffered DMEM without phenol red (containing 2 mM l-glutamine, and 1 mM sodium pyruvate) under basal conditions and in response to 10 mM glucose, 1 µM oligomycin, and 100 mM 2-deoxyglucose (Sigma-Aldrich, Burlington, MA, USA). The cells were analyzed with the Seahorse XFe-96 Extracellular Flux Analyzer (Agilent, Boulder, CO, USA). After plate reading, cells were carefully washed with PBS and lysed with 4 °C cooled Cytobuster protein extraction reagent (Merck-Millipore, Burlington, MA, USA) supplemented with protease inhibitor (Roche, Basilea, Switzerland). Then, proteins were quantified by the Bradford assay (ThermoFisher, Carlsbad, CA, USA) according to the manufacturer’s instructions. Each OCR and ECAR determination was normalized by the protein concentration of its respective well.

### 4.6. Ca^2+^ Measures

On 18 mm coverslips pre-coated with poly-L-lysine (Sigma-Aldrich, Burlington, MA, USA), 1 × 10^5^ cells were seeded to confluency. Next, the cells were pre-incubated with 5 μM XeB or the vehicle (control) for 60 min loaded with 5 μM of Fluo-4 AM and stimulated with 100 µM ATP. Confocal images were captured every 1.29 s. The experiments were performed in a Ca^2+^-free imaging medium contained DMEM without phenol red and high glucose.

### 4.7. Immunoblots

Cells were treated with 5 µM XeB or vehicle for 1 h and lysed on ice with Cytobuster protein extraction reagent (Merck-Millipore, Burlington, MA, USA) supplemented with protease and phosphatase inhibitors (complete PhosSTOP, Roche, Basilea, Switzerland). The protein quantification was performed by the Bradford assay (ThermoFisher, Carlsbad, CA, USA) according to the manufacturer’s instructions. Protein extracts were separated in 10% SDS-polyacrylamide gels, and transferred to PDVF membranes (Merck-Millipore, Burlington, MA, USA). Then, membranes were blocked in 5% fat-free milk for 1 h at room temperature and incubated overnight at 4 °C with primary antibody 1:1000 phospho-AMPKα Thr172 (Catalog #2531, Cell Signaling Technology, Danvers, MA, USA), 1:1000 rabbit antibody total AMPKα (Catalog #2532, Cell Signaling Technology, Danvers, MA, USA). Then, membranes were incubated for 1 h at room temperature with a secondary antibody conjugated to horseradish peroxidase (ThermoFisher, Carlsbad, CA, USA). Chemiluminescence detection and densitometric analyses were performed with ImageJ software, as described before [24].

### 4.8. NAD^+^/NADH Measures

The NAD^+^/NADH ratio was determined using a commercial kit (BioVision, K337-100) according to the manufacturer’s instructions. Briefly, 3 × 10^5^ cells were incubated either with 5 µM XeB or vehicle for 1 or 4 h. Cells were then washed twice with PBS, scraped off the dishes, and pelleted. NAD^+^ and NADH were extracted and the samples were subjected to two freeze–thaw cycles and centrifuged. Aliquots of each sample were heated at 60 °C for 30 min to decompose NAD^+^. Samples were then loaded into 96-well plates for absorbance measurements at 450 nm.

### 4.9. Cell Death

For cell death assays, 1 × 10^5^ cells were cultured and treated with the respective concentration of XeB. Cell death was determined by propidium iodide (PI, Thermo Fisher, Carlsbad, CA, USA) incorporation (5 mg/mL) by flow cytometry (FACS Aria^TM^ III, BD Biosciences, San José, CA, USA).

### 4.10. Statistics

All statistical analyses were performed using GraphPad Prism 4.03 (GraphPad Software, San Diego, CA, USA). The data are expressed as mean ± SEM of three independent experiments, each one performed in technical triplicates. Statistical analysis was performed using unpaired *t*-tests, one-way ANOVA with Bonferroni’s post hoc test for pairwise comparisons, or two-way ANOVA. The data were considered statistically significant at the 95% level (*p*  <  0.05).

## Figures and Tables

**Figure 1 ijms-22-00651-f001:**
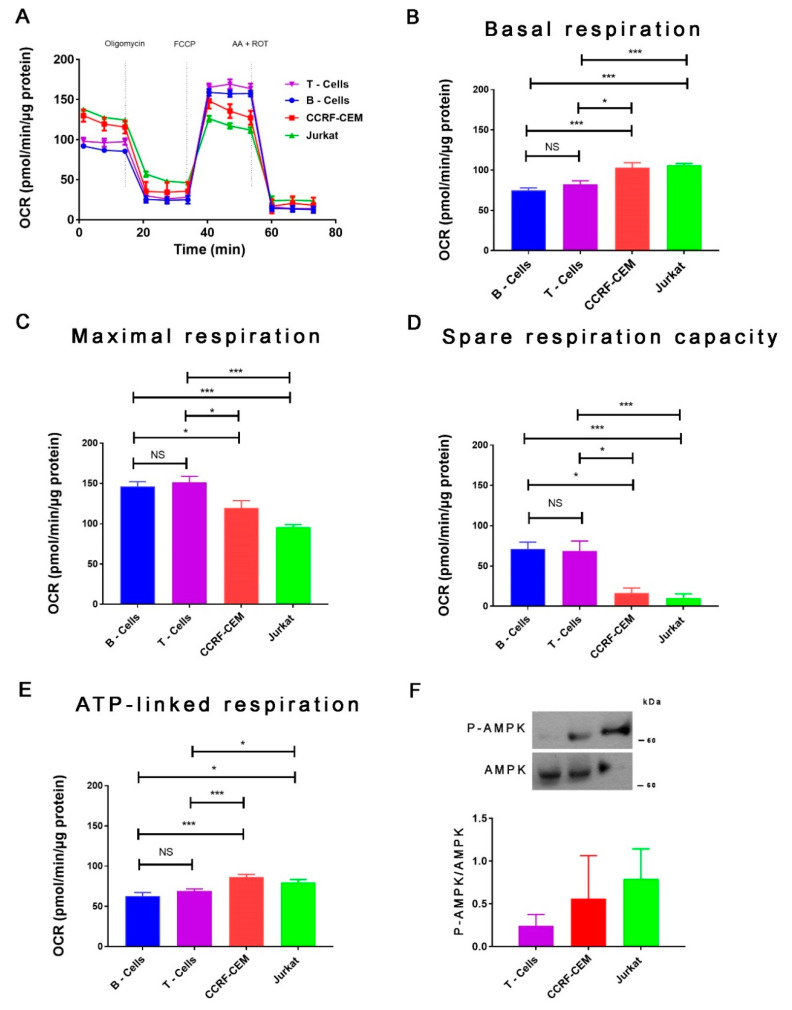
Mitochondrial respiratory function characterization in normal lymphocytes and T-ALL cell lines. (**A**) Oxygen consumption rates (OCR) in T-cells (purple), B-cells (blue), CCRF-CEM (red), and Jurkat (green) cells exposed sequentially to 1 µM oligomycin, 0.5 µM FCCP, and 1 µM rotenone (ROT) plus 1 µM antimycin A (AA). (**B**) Basal oxygen consumption rate (OCR) of T-cells (purple), B-cells (blue), CCRF-CEM (red), and Jurkat cells (green) calculated as the mean respiration rate before first injection minus the non-mitochondrial respiration rate. *n* = 6, mean ± S.E., * *p* < 0.05, *** *p* < 0.001, NS = not significant (*t*-test). (**C**) Maximal oxygen consumption rate (OCR) of T-cells (purple), B-cells (blue), CCRF-CEM (red), and Jurkat cells (green) calculated as the mean respiration rate measured after FCCP injection minus the non-mitochondrial respiration rate. *n* = 6, mean ± S.E., * *p* < 0.05, *** *p* < 0.001, NS = not significant (*t*-test). (**D**) Spare respiration capacity of T-cells (purple), B-cells (blue), CCRF-CEM (red), and Jurkat cells (green) calculated as the mean maximal respiration rate minus the mean basal respiration rate. *n* = 6, mean ± S.E., *** *p* < 0.001, NS = not significant (*t*-test). (**E**) ATP production linked respiration of T-cells (purple), B-cells (blue), CCRF-CEM (red), and Jurkat cells (green) calculated as the last rate measurement before oligomycin injection minus the minimum rate measurement after oligomycin injection. *n* = 6, mean ± S.E., * *p* < 0.05, *** *p* < 0.001 (*t*-test), NS = not significant. (**F**) Representative Western blot of phosphorylated AMPK (P-AMPK) and total AMPK (**top**) and P-AMPK/AMPK analysis (**bottom**) (*n* = 3).

**Figure 2 ijms-22-00651-f002:**
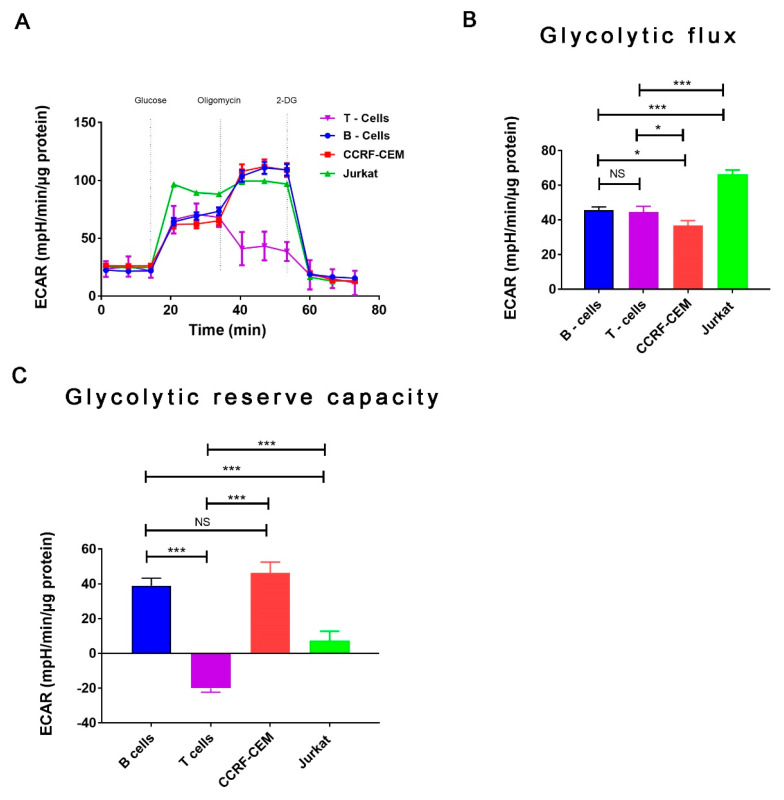
Glycolytic function characterization in normal lymphocytes and T-ALL cell lines. (**A**). Extracellular acidification rate (ECAR) in T-cells (purple), B-cells (blue), CCRF-CEM (red), and Jurkat cells (green) exposed sequentially to 10 mM glucose, 1 µM oligomycin, and 100 mM 2-deoxygucose (2-DG). (**B**). Glycolytic flux rate of T-cells (purple), B-cells (blue), CCRF-CEM (red), and Jurkat cells (green) calculated as the mean acidification rate after glucose injection minus the non-glycolytic acidification rate. *n* = 6, mean ± S.E., * *p* < 0.05, *** *p* < 0.001, NS = not significant (*t*-test). (**C**). Glycolytic reserve capacity of T-cells (purple), B-cells (blue), CCRF-CEM (red), and Jurkat cells (green) calculated as the mean acidification rate measured after oligomycin injection minus the mean acidification rate measured after glucose injection. *n* = 6, mean ± S.E., * *p* < 0.05, *** *p* < 0.001, NS = not significant (*t*-test).

**Figure 3 ijms-22-00651-f003:**
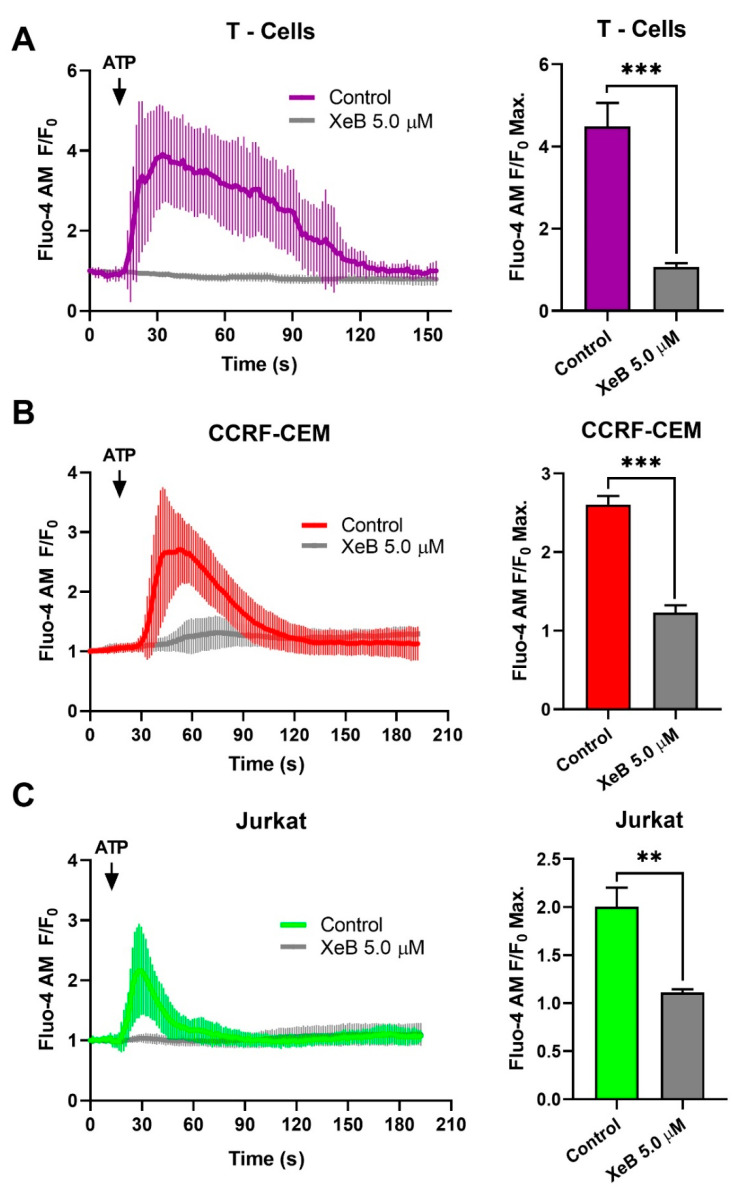
Xestospongin B inhibits InsP3R-mediated Ca^2+^ signaling in normal T cells and T-ALL cell lines. Left panels show representative traces of cytosolic Ca^2+^ and the right panels show the quantification (histogram) of maximal Fluo-4 AM fluorescence intensity in response to ATP stimulation (100 µM) in T-Cells (**A**), CCRF-CEM (**B**) and Jurkat (**C**) cells. Note the different time course of Ca^2+^-signals in response to ATP (arrow) in T-cells (A, purple trace), CCRF-CEM cells (B, red tracing) and Jurkat cells (C, green tracing), and the inhibition of Ca^2+^signals by 5 µM XeB (gray traces). Means ± S.D., of three independent experiments. ** *p* < 0.01 and *** *p* < 0.001.

**Figure 4 ijms-22-00651-f004:**
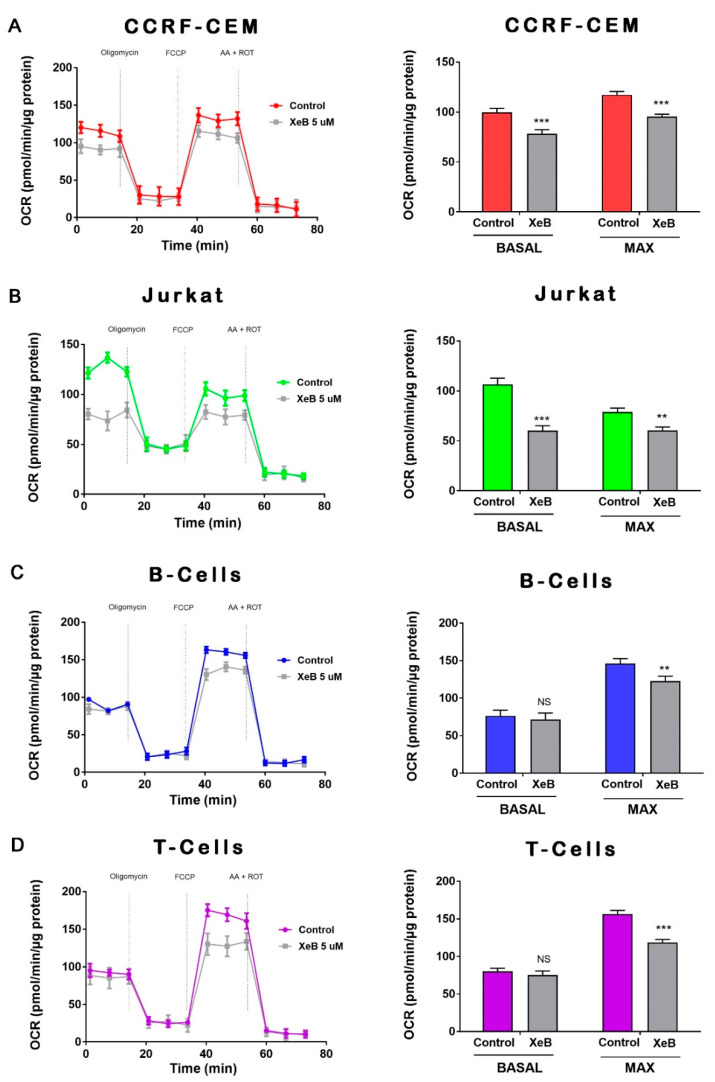
Effect of InsP3R inhibition on lymphocytes and T-ALL cell mitochondrial respiratory function. (**A**) Representative oxygen consumption traces (left), and basal and maximal oxygen consumption rate (OCR) analysis of CCRF-CEM control or treated with 5 µM XeB (right). (**B**) Representative oxygen consumption traces (left) and basal and maximal oxygen consumption rate (OCR) analysis of Jurkat control cells or those treated with 5 µM XeB (right). (**C**) Representative oxygen consumption traces (left) and basal and maximal oxygen consumption rate (OCR) analysis of control B cells or B cells treated with 5 µM XeB (right). (**D**). Representative oxygen consumption traces (left) and basal and maximal oxygen consumption rate (OCR) analysis of T-cells control or treated with 5 µM XeB (right). *n* = 3, mean ± S.E., ** *p* < 0.01, *** *p* < 0.001, NS = not significant (*t*-test).

**Figure 5 ijms-22-00651-f005:**
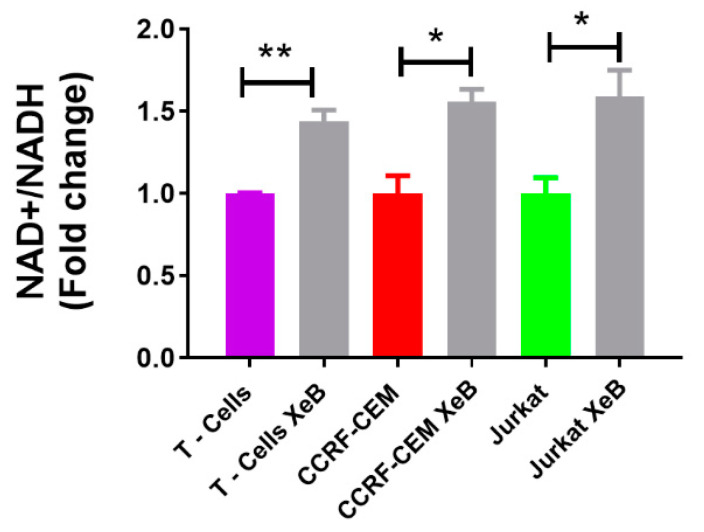
InsP3R inhibition increases NAD^+^/NADH ratio in normal and T-ALL cells. Analysis of the NAD^+^/NADH ratio in normal T cells, CCRF-CEM, and Jurkat cells treated with 5 µM XeB or the vehicle for 4 h. *n* = 3, mean ± S.E., * *p* < 0.05, ** *p* < 0.01 (*t*-test).

**Figure 6 ijms-22-00651-f006:**
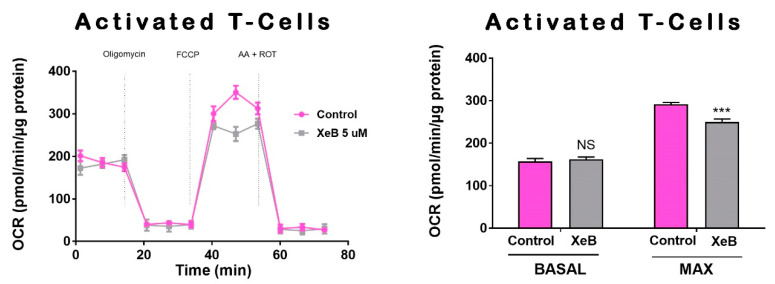
Effect of InsP3R inhibition on activated T-cells mitochondrial respiratory function. Representative oxygen consumption traces (**left**) and basal and maximal oxygen consumption rate (OCR) analysis of T-activated cell control or treated with 5 µM XeB (**right**). *n* = 3, mean ± S.E., *** *p* < 0.001, NS = not significant (*t*-test).

**Figure 7 ijms-22-00651-f007:**
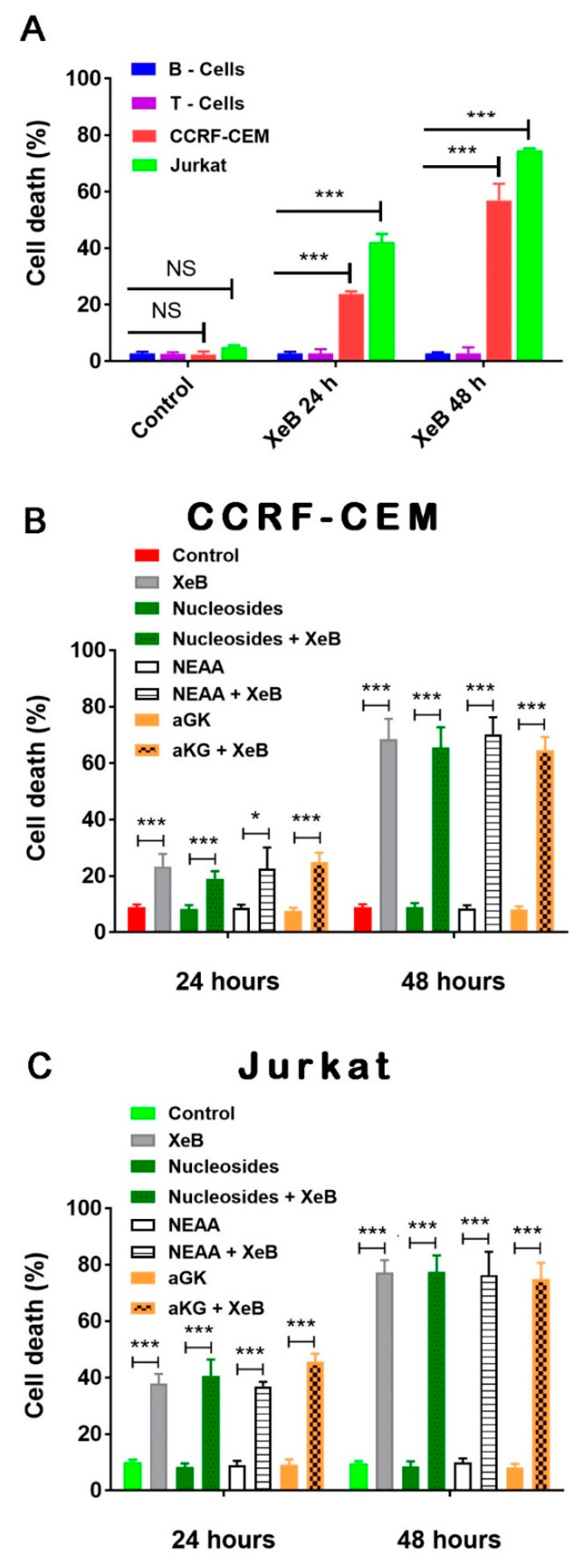
InsP3R inhibition induces cell death in T-ALL cells, which cannot be reverted by intermediate metabolites. (**A**) T-cells (purple), B-cells (blue), CCRF-CEM (red), and Jurkat (green) cells were treated with 5 μM XeB for 24 h and 48 h, and cell death was determined by propidium incorporation using flow cytometry. (**B**) CCRF-CEM cells were treated with 5 μM XeB for 24 and 48 h in the presence of either nucleosides, 5 mM α-ketoglutarate, or non-essential amino acids (NEAA) and cell death was determined by propidium incorporation using flow cytometry. (**C**) Jurkat cells were treated with 5 μM XeB for 24 and 48 h in the presence of either nucleosides, α-ketoglutarate, or NEAA, and cell death was determined by propidium incorporation by means of flow cytometry *n* = 3, mean ± S.E., * *p* < 0.05, *** *p* < 0.001 (*t*-test).

**Figure 8 ijms-22-00651-f008:**
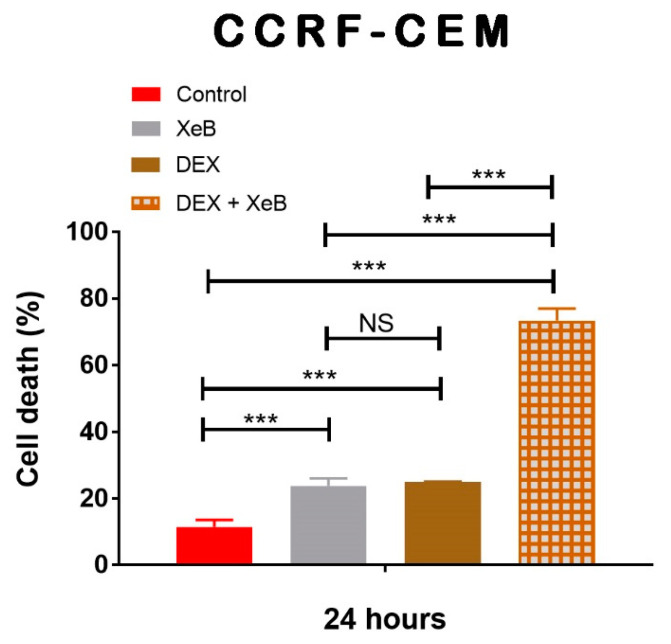
The inhibition of InsP3R with XeB enhances cell death in the presence of dexamethasone in CCRF-CEM cells. CCRF-CEM cells were treated with 5 μM XeB, 10 μM dexamethasone (DEX), or a combination of 5 μM XeB plus 10 μM dexamethasone (DEX) for 24 h and cell death was determined by propidium iodide incorporation by flow cytometry *n* = 3, mean ± S.E., *** *p* < 0.001 (*t*-test), NS = not significant (*t*-test).

**Figure 9 ijms-22-00651-f009:**
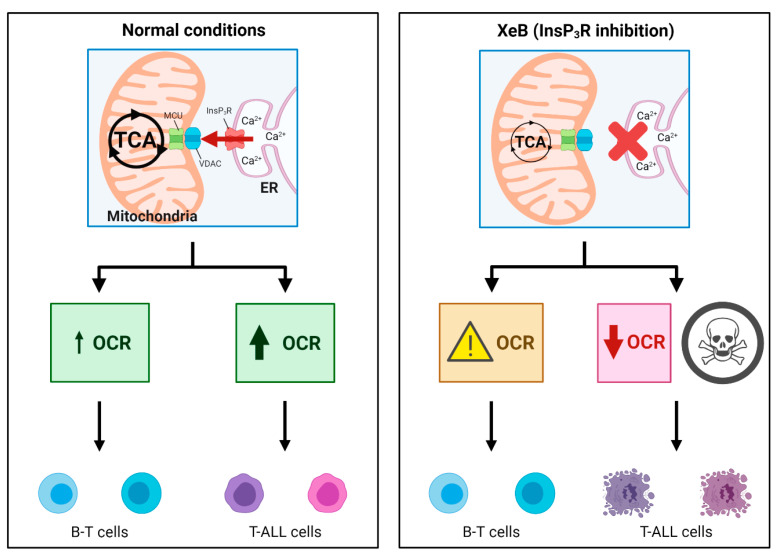
Proposed model for InsP3R-mitochondria communication and viability of T-ALL cells. Under normal conditions, InsP3R constitutively releases Ca^2+^ that feeds the mitochondria and stimulates the activity of essential enzymes of the TCA cycle, keeping a constant flux of ATP, reducing equivalents and intermediate metabolites in both normal B and T cells and T-ALL cells. The transfer of Ca^2+^ to mitochondria is especially important for T-ALL tumor cell lines, which show higher mitochondrial oxygen consumption rates (OCR) compared with non-malignant cells. Indeed, when InsP3R calcium release is specifically interrupted by XeB treatment, non-malignant cells display a slight decrease in maximal mitochondrial respiration, while T-ALL cells dramatically reduce their mitochondrial respiration and eventually die. This model was created with BioRender.com.

## Data Availability

The data presented in this study are available within the article.

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
