# Peer review of "Inhibition of InsP3R with Xestospongin B Reduces Mitochondrial Respiration and Induces Selective Cell Death in T Cell Acute Lymphoblastic Leukemia Cells"

_ijms, 2021, doi:10.3390/ijms22020651_

Round 1

Reviewer 1 Report

Corrections accepted.
Thank you for your interesting work.

Author Response

Thank for your feedback and support

Reviewer 2 Report

RE: Inhibition of InsP3R with Xestospongin B reduces mitochondrial respiration and induces selective cell death in T cell acute lymphoblastic leukemia cells by Pablo Cruz, Ulises Ahumada-Castro, Galdo Bustos, Jordi Molgó, Daniela Sauma, Alenka  Lovy and César Cárdenas

The authors present data showing the effects of InsP3R inhibition by XeB on mitochondria OCR, ECAR, and Ca2+ signaling. Comments are listed below in bullet format:

  • In the introduction, the author states that: “…increasing evidence indicates that most cancer cells rely on mitochondrial metabolism and use a significant fraction of glucose-derived pyruvate for ATP generation through oxidative phosphorylation (OXPHOS).”, which seems to establish the author’s narrative in which they conclude that T-ALL cells are more reliant on OXPHOS based on basal OCR and ATP-linked OCR in T-ALL cells.  However, references provided for this statement do not suggest that most cancer cells rely on glucose-derived pyruvate for ATP generation through OXPHOS.  On the contrary, one reference (pubmed ID: 25066121) provided by the author suggests that inhibition of pyruvate oxidation using the complex I inhibitor rotenone alternatively, promotes tumor migration and metastasis.  Can the author comment on this?

  • In a previous publication by the author (Pubmed ID:26947070), the addition of 5mM methyl-pyruvate restored colony formation in MCF7 cells when treated with XeB. Methyl-pyruvate is known to induce ROS formation (Pubmed ID: 24385472) in tumor cell lines. Because InsP3R inhibition by XeB affects matrix Ca2+ and enzyme activity, it may have a role in ROS mediation.  Has a similar experiment been conducted in T-ALL cells? If not, it would be interesting to test the effects of XeB on ROS production?

  • In Figure 1A, the authors present data related to basal and FCCP supported respiration in cells. For the FCCP addition, since only a single concentration was used, did the authors pre-determine the optimum FCCP required to elicit max respiration for each cell line? If not, then differences in calculated spare reserve capacity could be misinterpreted based on potential differences in FCCP sensitivity between the cell lines. If this was not done, then the authors should consider including a statement addressing this as a limitation.

  • Related to AMPK, the authors state ..”observed that in T-ALL cells, AMPK shows a clear tendency to be phosphorylated and therefore activate in basal conditions compared with normal T cells.”. Such a statement is not supported by the data in which pAMPK is not significantly difference between the groups. Either more experiments need to be performed or the language should likely be changed to reflect the lack of significance in the experiment.

  • Figure 2A and Figure 2C appear to display conflicting findings. The author states that “Glycolytic reserve capacity of T-Cells (purple), B-Cells (blue), CCRF-CEM (red) and Jurkat cells (green) calculated as the mean acidification rate measured after oligomycin injection minus the mean acidification rate measured after glucose injection.”.  However, Figure 2A shows a decrease of the ECAR after oligomycin addition to the T-cells indicating a negative glycolytic reserve capacity, while Figure 2C displays a glycolytic reserve capacity which reflects a positive value.  Likewise, the glycolytic reserve capacity of Jurkat cells appears to be much lower in Figure 2A than the calculated value displayed in Figure 2C.   Can the author specify which data points were used in the calculations?

  • Related to the ECAR measurement, the authors refer to this as a measurement of glycolysis. However, what is being measured is the acidification of the media, which can occur via glycolysis, as well as respiration (CO2) (PMID: 25449966). This does not appear to have been accounted for in the current paper. The authors should consider controlling for acidification due to CO2 or clearly state in the manuscript that the ECAR data, as displayed, is not a direct readout of glycolytic metabolism.

Author Response

Regarding our work entitled “Inhibition of InsP3R with Xestospongin B reduces mitochondrial respiration and induces selective cell death in T cell acute lymphoblastic leukemia cells” we thank the reviewers for their comments and the opportunity to improve our manuscript. We have revised the manuscript and addressed the issues that have been raised to the best of our abilities.

1-In the introduction, the author states that: “…increasing evidence indicates that most cancer cells rely on mitochondrial metabolism and use a significant fraction of glucose-derived pyruvate for ATP generation through oxidative phosphorylation (OXPHOS).”, which seems to establish the author’s narrative in which they conclude that T-ALL cells are more reliant on OXPHOS based on basal OCR and ATP-linked OCR in T-ALL cells.  However, references provided for this statement do not suggest that most cancer cells rely on glucose-derived pyruvate for ATP generation through OXPHOS.  On the contrary, one reference (pubmed ID: 25066121) provided by the author suggests that inhibition of pyruvate oxidation using the complex I inhibitor rotenone alternatively, promotes tumor migration and metastasis.  Can the author comment on this?

Response: We thank the reviewer for this comment. We agree that this is not the best reference for our statement, and we have eliminated it. Nevertheless, we take this opportunity to mention that cancer cells will reprogram their metabolism toward OXPHOS or glycolysis according to the tumor context, as recently reviewed by DeBerardinis Group (PMID: 32273439). Also, we added that emerging evidence from patient-derived tumors, which includes some forms of refractory leukemia such as AML, indicates that OXPHOS seems to play a main role in tumor cell survival, by providing not only ATP, but essential metabolic intermediates for the synthesis of biomolecules (PMID: 29892070). Please find in p2, lines 60-66, a brief discussion about this interesting topic.

2) In a previous publication by the author (Pubmed ID:26947070), the addition of 5mM methyl-pyruvate restored colony formation in MCF7 cells when treated with XeB. Methyl-pyruvate is known to induce ROS formation (Pubmed ID: 24385472) in tumor cell lines. Because InsP3R inhibition by XeB affects matrix Ca2+ and enzyme activity, it may have a role in ROS mediation.  Has a similar experiment been conducted in T-ALL cells? If not, it would be interesting to test the effects of XeB on ROS production?

Response: We thank the reviewer for this interesting question. In Cardenas et al, 2016 (Pubmed ID:26947070), we show that XeB does not induce ROS production in breast cancer cell lines. In addition, the treatment with N-acetyl-cystein (NAC) does not restore colony formation in MCF7 cells. Thus, we believe that ROS do not play a role in the effects mediated by InsP3R inhibition with XeB. However, we have not conducted similar experiments in T-ALL cells. Thus, we cannot discard a ROS increase in T-ALL cells after XeB application. Nevertheless, taking into consideration that the NAD+/NADH ratio increased, it is likely that the ROS production capacity is diminished. Future studies are necessary to prove this hypothesis. Please find in p15, lines 335-342 a brief discussion about this interesting topic.  

3)      In Figure 1A, the authors present data related to basal and FCCP supported respiration in cells. For the FCCP addition, since only a single concentration was used, did the authors pre-determine the optimum FCCP required to elicit max respiration for each cell line? If not, then differences in calculated spare reserve capacity could be misinterpreted based on potential differences in FCCP sensitivity between the cell lines. If this was not done, then the authors should consider including a statement addressing this as a limitation.

Response: This is a great comment from the reviewer. For this particular set of experiments, we used a FCCP concentration that we obtained after a concentration study we performed in T cells (PMID: 29472932). We agree with the reviewer that this could lead to sub-estimate the spare respiratory capacity. We include a statement addressing this limitation in p14, lines 324-327.

4)      Related to AMPK, the authors state ..”observed that in T-ALL cells, AMPK shows a clear tendency to be phosphorylated and therefore activate in basal conditions compared with normal T cells.”. Such a statement is not supported by the data in which pAMPK is not significantly difference between the groups. Either more experiments need to be performed or the language should likely be changed to reflect the lack of significance in the experiment.

Response: We agree with reviewer. We would like to do more experiments to strengthen this point but we are not allowed in the lab because of COVID-19 regulations. Thus, we have corrected the manuscript accordingly. In p3, lines 105-107 you will find the change.

5)      Figure 2A and Figure 2C appear to display conflicting findings. The author states that “Glycolytic reserve capacity of T-Cells (purple), B-Cells (blue), CCRF-CEM (red) and Jurkat cells (green) calculated as the mean acidification rate measured after oligomycin injection minus the mean acidification rate measured after glucose injection.”.  However, Figure 2A shows a decrease of the ECAR after oligomycin addition to the T-cells indicating a negative glycolytic reserve capacity, while Figure 2C displays a glycolytic reserve capacity which reflects a positive value.  Likewise, the glycolytic reserve capacity of Jurkat cells appears to be much lower in Figure 2A than the calculated value displayed in Figure 2C.   Can the author specify which data points were used in the calculations?

Response: We really appreciate that the reviewer points this out. I apologize but the bar graph in 2C does not represent the data of figure 2A. Somehow the wrong bar graph was inserted into the manuscript. The correct analysis is present now as figure 2C and as suggested by the reviewer the numbers for T cells are negative, which tells us that they don’t have glycolytic reserve capacity. Also, as pointed out by the reviewer, the glycolytic reserve in Jurkat cells is small. We have modified the text accordingly in p3, lines 112-115. Nevertheless, this does not change the main message of our manuscript.

Regarding which data points in Figure 2 were used in the calculations, we used the third point after glucose injection and the third point after oligomycin injection for our analysis.

6)      Related to the ECAR measurement, the authors refer to this as a measurement of glycolysis. However, what is being measured is the acidification of the media, which can occur via glycolysis, as well as respiration (CO2) (PMID: 25449966). This does not appear to have been accounted for in the current paper. The authors should consider controlling for acidification due to CO2 or clearly state in the manuscript that the ECAR data, as displayed, is not a direct readout of glycolytic metabolism.

Response: We agree with the reviewer that respiration also contributes to the acidification of the media. Controlling CO2 acidification of the media is a great idea that we plan to implement in the near future. Unfortunately, right now it is impossible to go to the lab and do any kinds of experiments.

We now mention in the manuscript (p14, lines 287-293) that CO2 from respiration contributes to the acidification of the media, and the range of this contribution can vary from 0% to 100% from one cell type to another.

This manuscript is a resubmission of an earlier submission. The following is a list of the peer review reports and author responses from that submission.

Round 1

Reviewer 1 Report

The subject is interesting and experiments are well carried out, although limited to Seahorse and cell viability experiments and being largely confirmatory.

Major concerns:

It is not quite obvious to me how this manuscript contributes to the field. Two cancer cell lines are studied and compared to B- and T-cells. One type of pharmacological manipulation is used. The results are of confirmatory nature. Previous studies have explored many of the aspects studies here and have utilized knock-down strategies. It is not quite clear to me how the experiments presented here (if so, this is poorly explained).

I am a bit confused by the Discussion. It is very short (less than one page) and is mainly a summary of the literature. The results of the study are not extensively discussed – except that they confirm what has already been published.

Minor concerns:

I would have had expected a larger difference between normal B/T-cells and these proliferating cancer cells than what is observed here. Having some experience with the Seahorse I wonder about how experiments are standardized. Authors use OCR/ug protein. How different would the results be is expressed per cell? 

In Fig 3 cells were treated with XeB. Were cells pretreated? An advantage with the Seahorse is that drugs are injected during the experiment, allowing a baseline to be set. I would have prefered having the cells plated first, the recording started and the XeB then injected. Does XeB affect the baseline OCR under these conditons?. How rapid is the effect?

Author Response

Regarding our work entitled “Inhibition of InsP3R with Xestospongin B reduces mitochondrial respiration and induces selective cell death in T cell acute lymphoblastic leukemia cells” we thank the reviewers for their comments and the opportunity to improve our manuscript. We have revised the manuscript and addressed the issues that have been raised to the best of our abilities.

Reviewer comments:

Reviewer #1

The subject is interesting and experiments are well carried out, although limited to Seahorse and cell viability experiments and being largely confirmatory.

Response: We appreciate that the reviewer finds our work interesting and well carried out.

Major concerns:

It is not quite obvious to me how this manuscript contributes to the field. Two cancer cell lines are studied and compared to B- and T-cells. One type of pharmacological manipulation is used. The results are of confirmatory nature. Previous studies have explored many of the aspects studies here and have utilized knock-down strategies. It is not quite clear to me how the experiments presented here (if so, this is poorly explained).

Response: The role of the of the InsP3R in the context of cancer has just begun to be understood. Our group demonstrated that constitutive calcium transfer to the mitochondria mediated by the InsP3R is essential to maintain mitochondrial function. Molecular or pharmacological inhibition of the InsP3R causes a bioenergetic crisis that selectively kills cancer (breast and prostate) cells (Cardenas et al., 2016). More recently we have shown that cancer cells which don’t use mitochondria as a bioenergetic hub (pheochromocytoma, hypoxic cells, etc.) are also affected (Cardenas et al., 2020). We demonstrated that these cells still require calcium transfer to the mitochondria to maintain the activity of the alpha-ketoglutarate dehydrogenase, which is fundamental to maintain cellular metabolism through reductive carboxylation. Unpublished data from our lab which belongs to another manuscript shows that triple negative breast cancer cells are resistant to the treatment with XeB (Supplementary figure 1 for the reviwer), indicating that every cell type may behave differently. Thus, although our results seem to be confirmatory, without the experimental evidence we could not have predicted the outcome in T-ALL cells.

In leukemia, lymphoma and other cancer types that affect the white blood linage, the role of InsP3R has been study mostly in the context of an apoptosis mediator. In these cancer types, Distelhorst’s and Bultynck’s groups have shown that InsP3R activity is reduced to avoid overwhelming mitochondrial calcium levels and apoptosis. Along these lines, activation of the InsP3R receptors induces cells death. Although reduced, the activity of the InsP3R was still present in these cells. Its physiological role is still unknown. We believe the remaining activity of the InsP3R is necessary to sustain mitochondrial function and therefore bioenergetic and metabolic homeostasis and its inhibition will cause cell death. A previous report showed that the expression of ORP4L in T-ALL favors InsP3 generation and therefore InsP3R activation and bioenergetics (Zhong et al., 2016). In this work molecular tools where used to manipulate the expression of ORP4L, not the InsP3R. In fact, related to the control of bioenergetics and metabolism by InsP3R, no studies using knock-down strategies have been performed in T-ALL cell or in other blood cancer diseases. Moreover, this is the first time that the inhibition of the InsP3R is shown to affect bioenergetics of normal and T-ALL cells and actually causing cell death in the former.

We chose to work with CCRF-CEM cells because it is a well described model of T-ALL and represents fairly well the behavior of T-ALL cancer cells. The Jurkat also are a well-established cellular model studied in the context of T-ALL that have the particularity of being insensitive to glucocorticoids, a phenomenon observed in patients. Thus, we believe these two cell lines are good models of the disease. As controls, the use of T and B cells helps address whether our pharmacological approach will affect the viability of normal cells, which we prove not to be the case. It is important to highlight that this is the first report demonstrating that InsP3R may have a therapeutic potential in T-ALL.

In previous work we have shown that the knockdown of the InsP3Rs exhibits the same effect as the inhibition of InsP3R with the specific inhibitor Xestospongine B (XeB). XeB is a natural compound with a known structure. Unfortunately, the availability of this compound is scarce.

By focusing this work on XeB, we want to raise awareness about the importance of this compound as a therapeutic alternative for the treatment of cancer and encourage the scientific community to generate similar compounds that allow the field to move forward. 

I am a bit confused by the Discussion. It is very short (less than one page) and is mainly a summary of the literature. The results of the study are not extensively discussed – except that they confirm what has already been published.

Response: As a communication we try to keep every part of the manuscript short, but with no doubt the discussion is missing some information. We appreciate this comment. We have added our results to the discussion and developed it a little more.

Minor concerns:

I would have had expected a larger difference between normal B/T-cells and these proliferating cancer cells than what is observed here. Having some experience with the Seahorse I wonder about how experiments are standardized. Authors use OCR/ug protein. How different would the results be is expressed per cell?

Response: We thanks the reviewer for this comment. We seed the same number of cells per well for each of the cell types used in this manuscript, but we still normalize by protein because it helps to eliminate variation caused by the initial pipetting. Nevertheless, if we normalized the OCR data shown in this manuscript by cell number the recording and the differences remain the same.  

In Fig 3 cells were treated with XeB. Were cells pretreated? An advantage with the Seahorse is that drugs are injected during the experiment, allowing a baseline to be set. I would have prefered having the cells plated first, the recording started and the XeB then injected. Does XeB affect the baseline OCR under these conditons?. How rapid is the effect?

Response: All the seahorse experiments shown in this manuscript were done in the presence of XeB after been pretreated for 1h. Injecting the compound is an interesting idea that we have tried before in other cell types.

Using a similar protocol as the one used in this manuscript, we didn’t observe a reduction in the basal respiration after injecting XeB. However, the maximum respiration did show a little drop. We believe that although the inhibition of the InsP3R with XeB happens very fast (after 5 min no calcium signals are observed), the effect on respiration takes longer to display.

Encouraged by the reviewer’s comment, we are planning to do an experiment (as soon we can come back to the lab, covid-19 protocol does not allow us in the lab yet) in which we will record basal oxygen consumption and then inject XeB and record respiration for over an hour.

Nevertheless, the results from this experiment won’t change the main conclusion of the present work, which is that InsP3Rs are essential to maintain mitochondrial function and the viability of T-ALL cells.

Regarding our work entitled “Inhibition of InsP3R with Xestospongin B reduces mitochondrial respiration and induces selective cell death in T cell acute lymphoblastic leukemia cells” we thank the reviewers for their comments and the opportunity to improve our manuscript. We have revised the manuscript and addressed the issues that have been raised to the best of our abilities.

Reviewer comments:

Reviewer #1

The subject is interesting and experiments are well carried out, although limited to Seahorse and cell viability experiments and being largely confirmatory.

Response: We appreciate that the reviewer finds our work interesting and well carried out.

Major concerns:

It is not quite obvious to me how this manuscript contributes to the field. Two cancer cell lines are studied and compared to B- and T-cells. One type of pharmacological manipulation is used. The results are of confirmatory nature. Previous studies have explored many of the aspects studies here and have utilized knock-down strategies. It is not quite clear to me how the experiments presented here (if so, this is poorly explained).

Response: The role of the of the InsP3R in the context of cancer has just begun to be understood. Our group demonstrated that constitutive calcium transfer to the mitochondria mediated by the InsP3R is essential to maintain mitochondrial function. Molecular or pharmacological inhibition of the InsP3R causes a bioenergetic crisis that selectively kills cancer (breast and prostate) cells (Cardenas et al., 2016). More recently we have shown that cancer cells which don’t use mitochondria as a bioenergetic hub (pheochromocytoma, hypoxic cells, etc.) are also affected (Cardenas et al., 2020). We demonstrated that these cells still require calcium transfer to the mitochondria to maintain the activity of the alpha-ketoglutarate dehydrogenase, which is fundamental to maintain cellular metabolism through reductive carboxylation. Unpublished data from our lab which belongs to another manuscript shows that triple negative breast cancer cells are resistant to the treatment with XeB (Supplementary figure 1 for the reviwer), indicating that every cell type may behave differently. Thus, although our results seem to be confirmatory, without the experimental evidence we could not have predicted the outcome in T-ALL cells.

In leukemia, lymphoma and other cancer types that affect the white blood linage, the role of InsP3R has been study mostly in the context of an apoptosis mediator. In these cancer types, Distelhorst’s and Bultynck’s groups have shown that InsP3R activity is reduced to avoid overwhelming mitochondrial calcium levels and apoptosis. Along these lines, activation of the InsP3R receptors induces cells death. Although reduced, the activity of the InsP3R was still present in these cells. Its physiological role is still unknown. We believe the remaining activity of the InsP3R is necessary to sustain mitochondrial function and therefore bioenergetic and metabolic homeostasis and its inhibition will cause cell death. A previous report showed that the expression of ORP4L in T-ALL favors InsP3 generation and therefore InsP3R activation and bioenergetics (Zhong et al., 2016). In this work molecular tools where used to manipulate the expression of ORP4L, not the InsP3R. In fact, related to the control of bioenergetics and metabolism by InsP3R, no studies using knock-down strategies have been performed in T-ALL cell or in other blood cancer diseases. Moreover, this is the first time that the inhibition of the InsP3R is shown to affect bioenergetics of normal and T-ALL cells and actually causing cell death in the former.

We chose to work with CCRF-CEM cells because it is a well described model of T-ALL and represents fairly well the behavior of T-ALL cancer cells. The Jurkat also are a well-established cellular model studied in the context of T-ALL that have the particularity of being insensitive to glucocorticoids, a phenomenon observed in patients. Thus, we believe these two cell lines are good models of the disease. As controls, the use of T and B cells helps address whether our pharmacological approach will affect the viability of normal cells, which we prove not to be the case. It is important to highlight that this is the first report demonstrating that InsP3R may have a therapeutic potential in T-ALL.

In previous work we have shown that the knockdown of the InsP3Rs exhibits the same effect as the inhibition of InsP3R with the specific inhibitor Xestospongine B (XeB). XeB is a natural compound with a known structure. Unfortunately, the availability of this compound is scarce.

By focusing this work on XeB, we want to raise awareness about the importance of this compound as a therapeutic alternative for the treatment of cancer and encourage the scientific community to generate similar compounds that allow the field to move forward. 

I am a bit confused by the Discussion. It is very short (less than one page) and is mainly a summary of the literature. The results of the study are not extensively discussed – except that they confirm what has already been published.

Response: As a communication we try to keep every part of the manuscript short, but with no doubt the discussion is missing some information. We appreciate this comment. We have added our results to the discussion and developed it a little more.

Minor concerns:

I would have had expected a larger difference between normal B/T-cells and these proliferating cancer cells than what is observed here. Having some experience with the Seahorse I wonder about how experiments are standardized. Authors use OCR/ug protein. How different would the results be is expressed per cell?

Response: We thanks the reviewer for this comment. We seed the same number of cells per well for each of the cell types used in this manuscript, but we still normalize by protein because it helps to eliminate variation caused by the initial pipetting. Nevertheless, if we normalized the OCR data shown in this manuscript by cell number the recording and the differences remain the same.  

In Fig 3 cells were treated with XeB. Were cells pretreated? An advantage with the Seahorse is that drugs are injected during the experiment, allowing a baseline to be set. I would have prefered having the cells plated first, the recording started and the XeB then injected. Does XeB affect the baseline OCR under these conditons?. How rapid is the effect?

Response: All the seahorse experiments shown in this manuscript were done in the presence of XeB after been pretreated for 1h. Injecting the compound is an interesting idea that we have tried before in other cell types.

Using a similar protocol as the one used in this manuscript, we didn’t observe a reduction in the basal respiration after injecting XeB. However, the maximum respiration did show a little drop. We believe that although the inhibition of the InsP3R with XeB happens very fast (after 5 min no calcium signals are observed), the effect on respiration takes longer to display.

Encouraged by the reviewer’s comment, we are planning to do an experiment (as soon we can come back to the lab, covid-19 protocol does not allow us in the lab yet) in which we will record basal oxygen consumption and then inject XeB and record respiration for over an hour.

Nevertheless, the results from this experiment won’t change the main conclusion of the present work, which is that InsP3Rs are essential to maintain mitochondrial function and the viability of T-ALL cells.

Reviewer 2 Report

Based on the observations described in the current manuscript authors conclude that T-ALL cell lines present exhibit higher mitochondrial respiration than non- malignant cells. Autors show that prolonged treatment with XeB causes cell death only malignant cells without affecting the normal counterpart and speculate that XeB may have application as a potential therapeutic agent in leukemia treatment.

Article is clearly written and based on sufficient and statistically significant experimental data, which illustrated by well-prepared and described figures. Methodological approach demonstrate the importance of the scientific topic of current study and selection of appropriate methods of testing of XeB as well as appropriate cells cultures. The presented manuscript is logical and clear, the authors ' vision and the presented results inspire confidence.

Comments:

1) for reliable use of the term «synergic effect», it is necessary to use Chou-Talalay combination index (CI);

2) correct the term «cytotoxicity» (figure 5 and 6)

Author Response

Reviewer #2

Based on the observations described in the current manuscript authors conclude that T-ALL cell lines present exhibit higher mitochondrial respiration than non- malignant cells. Autors show that prolonged treatment with XeB causes cell death only malignant cells without affecting the normal counterpart and speculate that XeB may have application as a potential therapeutic agent in leukemia treatment.

Article is clearly written and based on sufficient and statistically significant experimental data, which illustrated by well-prepared and described figures. Methodological approach demonstrate the importance of the scientific topic of current study and selection of appropriate methods of testing of XeB as well as appropriate cells cultures. The presented manuscript is logical and clear, the authors ' vision and the presented results inspire confidence.

Response: We thank the reviewer for the kind comments.

Comments:

  • for reliable use of the term «synergic effect», it is necessary to use Chou-Talalay combination index (CI).

Response: We apologize for the wrong use of the term “synergic effect”. To properly use it, as the reviewer mentioned, it is necessary to use the Chou-Talalay combination index, which requires the IC50 of XeB which we didn’t calculate for this particular cell line. We used a single concentration based on our previous experience with other cell lines. Thus, we have eliminated the term synergy from our manuscript.

2) correct the term «cytotoxicity» (figure 5 and 6)

Response: As requested by the reviewer we have changed the term cytotoxicity for cell death.

Reviewer 3 Report

To the authors The article by Cruz et al. focus on the inhibition of the transfer of calcium from the endoplasmic reticulum to the mitochondria using a previously published InsP3R inhibitor. The work mainly focusses on functional measurements using Extracellular Flux Analyzer and evaluation of cytotoxicity in two different characterized T-ALL cell lines. The OCR measurements are accurate, the normalization to protein amount adequate. However, essential mechanistic questions are not addressed. In summary, I have to raise major and some minor concerns before considering publication in IJMS.

Major remarks

  • The authors show functional measurements (OCR, ECAR, ATP-linked respiration), stating that “[…] cells have higher levels than B and T cells”. These data should be supported by specific ATP measurements, including e.g. the ADP/ATP-ratio.
  • The authors must show evidence that the inhibitor inhibits calcium transfer to the mitochondria, e.g. by using the Ca2+ -indicator Fluo-4 AM.
  • The authors should address the reasons for changes in the glycolysis level: which mechanism could be held responsible?
  • The authors should discuss the benefit of further investigation the respiratory chain being responsible for OCR.

Minor remarks

  • The authors should describe the method of protein normalization in the methods section as it is essential in OCR measurements.
  • The authors should describe the number of cells for cell death assessment.
  • Fig. 1: the axes across figure 1 should be scaled the same way
  • the axes label should be consistent in all figures (e.g. “B-Cells” vs “B cells”)
  • Fig. 5: a bar should indicate the significance indication as it is not clearly arranged, yet. · Fig. 6: the y-axis is labeled twice.
  • The manuscript would dramatically benefit from proofreading (e.g. l 93 “founnd”, l. 257 “evidenc”).
  • Even, glucocorticoid treatment is the cornerstone in T-ALL therapy, the authors may consider using more recent references for treatment of T-ALL (see reference 2).

Author Response

Responses to Reviewers’ Comments

Regarding our work entitled “Inhibition of InsP3R with Xestospongin B reduces mitochondrial respiration and induces selective cell death in T cell acute lymphoblastic leukemia cells” we thank the reviewers for their comments and the opportunity to improve our manuscript. We have revised the manuscript and addressed the issues that have been raised to the best of our abilities.

The article by Cruz et al. focuses on the inhibition of the transfer of calcium from the endoplasmic reticulum to the mitochondria using a previously published InsP3R inhibitor. The work mainly focuses on functional measurements using Extracellular Flux Analyzer and evaluation of cytotoxicity in two different characterized T-ALL cell lines. The OCR measurements are accurate, the normalization to protein amount adequate. However, essential mechanistic questions are not addressed. In summary, I have to raise major and some minor concerns before considering publication in IJMS.

Response: We appreciate that the reviewer finds our OCR data accurate and the normalization of protein adequate. We understand that our work has some mechanistic limitations. We have added new data that we believe strengthens our work.

Major remarks 

The authors show functional measurements (OCR, ECAR, ATP-linked respiration), stating that “[…] cells have higher levels than B and T cells”. These data should be supported by specific ATP measurements, including e.g. the ADP/ATP-ratio.

Response: The phrase “[…] cells have higher levels than B and T cells” corresponds specifically to the analysis of the ATP-linked respiration in these cells and is not a general conclusion of their full respiration profile. We understand that this was not very clear and we have changed it. We have added a general conclusion that points out that the T-ALL cells CCRF-CEM and Jurkat are under a bioenergetic stress.

The new text reads as follows;

“The OCR link to ATP production shows that the high levels of basal oxygen consumption in CCRF-CEM and Jurkat cells is used to generate ATP (Figure 1E). Altogether, these results show that T-ALL cell lines CCRF-CEM and Jurkat are using mitochondria to the maximum of their capability to generate ATP and maintain their homeostasis”.

The reviewer suggests supporting our data by adding for example the ADP/ATP ratio. We agree that more data was necessary to complement our results. However, we considered that the ADP/ATP ratio is not the best measurement since it tends to be quite stable thanks to the high activity of the adenylate cyclase (Hardie, 2003). Nevertheless, to get a sense of what is happening with the ATP levels, we determined the phosphorylation status of AMPK, which increases when the AMP/ATP ratio increases. As is shown in the new figure 1F, T-ALL cells show a clear tendency toward higher levels of phosphorylation, which indicate that the levels of ATP in these cells are not enough to maintain normal homeostasis and AMPK is active in response to this energetic stress.  Unfortunately, we were not able to repeat this experiment to reach statistical significance, given that the universities in Chile are closed because of the covid-19 pandemic. Nevertheless, we believe the tendency shown by AMPK supports our data. In addition, we determined the NAD+/NADH ratio which reflects the metabolic, redox and energetic status of the cells and which we have recently used successfully (Cardenas et al., 2020, Lovy et al., 2020) to determine metabolic stress induced by a decrease in mitochondrial function. As is shown in the new figure 5, the inhibition of InsP3R with XeB increases the NAD+/NADH ratio, which confirms the generation of metabolic stress and supports our OCR data.  

References.

Minireview: The AMP-Activated Protein Kinase Cascade: The Key Sensor of Cellular Energy Status D. Grahame Hardie

Endocrinology, Volume 144, Issue 12, 2003, Pages 5179–5183,

Cancer cells with defective oxidative phosphorylation require endoplasmic reticulum-to-mitochondria Ca2+ transfer for survival.

Cardenas C, Lovy A, Silva-Pavez E, Urra F, Mizzoni C, Ahumada-Castro U, Bustos G, Jaňa F, Cruz P, Farias P, Mendoza E, Huerta H, Murgas P, Hunter M, Rios M, Cerda O, Georgakoudi I, Zakarian A, Molgó J, Foskett JK.

Sci Signal. 2020;13(640):eaay1212.

Concerted Action of AMPK and Sirtuin-1 Induces Mitochondrial Fragmentation Upon Inhibition of Ca2+ Transfer to Mitochondria.

Lovy A, Ahumada-Castro U, Bustos G, Farias P, Gonzalez-Billault C, Molgó J, Cardenas C.

Front Cell Dev Biol. 2020 May 25;8:378.

The authors must show evidence that the inhibitor inhibits calcium transfer to the mitochondria, e.g. by using the Ca2+ -indicator Fluo-4 AM.

Response: As suggested by the reviewer, we determined the InsP3R-mediated Ca2+ release induced by ATP with Fluo-4 AM in T normal cells, CCRF-CEM and Jurkat cells in the presence of 5µM XeB. As expected, XeB almost completely inhibited the release of calcium mediated by InsP3R. The data has been incorporated as the new figure 3.

The authors should address the reasons for changes in the glycolysis level: which mechanism could be held responsible?

Response: The glycolytic measurement in our work follows the same patterns that have already been described for normal B cells, Jurkat and CCRF-CEM cells. We believe the reviewer is asking for the behavior of normal naïve T cells, which is expected to rely mostly on mitochondria, however, shows high levels of glycolysis. We believe this is the result of the Crabtree effect, which describes that OXPHOS is inhibited when high levels of glucose are available in the media, which is the case, since we culture our normal naïve T cells in high glucose (25mM) DMEM. We have discussed this in the text, which has been amended as follows;

Glycolysis has been described to be higher in T-ALL cells compared with naïve T cells. In particular, Jurkat and CCRF-CEM cells present a high energetic profile characterized by high glycolysis and high OXPHOS, as we observe here. Normal naïve T cells are expected to have high OXPHOS, as observed here, and low glycolysis, however, in our hands, glycolytic flux in these cells was high and the glycolytic reserve capacity low accordingly. We believe this metabolic behavior corresponds to a Crabtree effect, which describes that OXPHOS is inhibited by higher concentrations of glucose in the media such as the one we used in our work (25 mM).

The authors should discuss the benefit of further investigation the respiratory chain being responsible for OCR.

Response: Today we know that many cancers, not only hematological, rely on mitochondrial function for their survival, which makes research of the respiratory chain and other mitochondrial components that affect respiration for the development of agents with antitumoral action essential. As suggested by the reviewer, we highlight this throughout the manuscript. Specifically, in the discussion section we give some examples of cancer types where OXPHOS had been found to be upregulated and would benefit from the development of agents that affect respiration.

The text now reads as follows;

Increases in OXPHOS have been described in other hematological malignancies such as acute myelogenous leukemia (AML), chronic lymphocyte leukemia (CLL) and diffuse large B cell lymphoma as well as many other type of cancer such as pancreatic, lung adenocarcinoma, endometrial carcinoma, melanoma and glioma.   This information has made the discovery and development of inhibitors that affect OXPHOS function, such as XeB, an essential task.

Minor remarks.

The authors should describe the method of protein normalization in the methods section as it is essential in OCR measurements.

Response: We have added a description of the method used for protein normalization in the materials and methods section 4.5- bioenergetic profile.

  • The authors should describe the number of cells for cell death assessment.

Response: For cell death assays, 1 x 105 cells were cultured and treated with the respective concentration of XeB. This information has been added to the materials and methods section.

  • Fig. 1: the axes across figure 1 should be scaled the same way

Response: We have made the adjustments that the reviewer suggested.

  • the axes label should be consistent in all figures (e.g. “B-Cells” vs “B cells”)

Response: We are sorry for this mistake; all the axes are now labeled as B-Cells

  • Fig. 5: a bar should indicate the significance indication as it is not clearly arranged, yet.

Response: We have done what the reviewer suggested.

  • Fig. 6: the y-axis is labeled twice.

Response: We are sorry. We have fixed this mistake.

  • The manuscript would dramatically benefit from proofreading (e.g. l 93 “founnd”, l. 257 “evidenc”).

Response: We apologize for these mistakes. We have carefully read the manuscript and amended these mistakes. Also, an english native speaker has read our work and helped us with the grammar of the article.

  • Even, glucocorticoid treatment is the cornerstone in T-ALL therapy, the authors may consider using more recent references for treatment of T-ALL (see reference 2)

Response: We thank the reviewer for this comment. As suggested, we mention in the introduction that small molecule inhibitors of several intracellular pathways as well as immunotherapeutic approaches are being developed to treat refractory T-ALL malignancy. However, since 2005, no new treatment has been approved for the FDA to treat T-ALL, which strongly suggests that  a better understanding of the biology of T-ALL cells is necessary to be able to generate new molecular and pharmacological tools that can lead to new therapeutic options.

The new text added reads as follows;

there still exists a high percentage of primary and secondary resistance, which has encouraged the search for new agents and strategies to treat this disease. Small molecule inhibitors such as venetoclax (Bcl-2 inhibitor), dasatinib (tyrosine kinase inhibitor) and palbociclib (cyclin dependent kinases 4 and 6 inhibitor) have been used with certain success. Immunotherapeutic approaches with chimeric antigen receptor (CAR)-T and monoclonal antibodies have also shown encouraging results. Despite these efforts, no new agents for the treatment of relapsed and/or refractory T-ALL in adults and children have been approved for the FDA since nelarabine in 2005. Understanding the unique features of these cells is necessary to bring new options for treatment.